# A New Approach to Preform Design in Metal Forging Processes Based on the Convolution Neural Network

**Seungro Lee** [1],[†] , **Luca Quagliato** [1],[†] , **Donghwi Park** [1] , **Inwoo Kwon** [2] , **Juhyun Sun** [2] and **Naksoo Kim** [1],*

1   Department of Mechanical Engineering, Sogang University, Seoul 04107, Korea; lsr1310@sogang.ac.kr (S.L.); lucaq@sogang.ac.kr (L.Q.); pdhwi93@sogang.ac.kr (D.P.)
2   Research Department, Dong Yang Piston Co., Ltd., Ansan-si 15420, Gyeonggi do, Korea; iwkwon@dypiston.co.kr (I.K.); jhsun@dypiston.co.kr (J.S.)
*   Correspondence: nskim@sogang.ac.kr; Tel.: +82-27038635
†   These authors contributed equally to this work.

**Abstract:** This study presents an innovative methodology for preform design in metal forging processes based on the convolution neural network (CNN) algorithm. The proposed approach extracts the features of inputted forging product geometries and utilizes them to derive the corresponding preform shapes by employing weight arrays (filters) determined during the convolutional operations. The filters are progressively updated during the training process, emulating the learning steps of a process engineer responsible for the design of preform shapes for the forging processes. The design system is composed of multiple three-dimensional (3D) CNN sub-models, which can automatically derive individual 3D preform design candidates. It also implies that the 3D surfaces of preforms are easily acquired, which is important for the forging industry. The proposed preform design methodology was validated by applying it to two-dimensional (2D) axisymmetric shapes, one-quarter plane-symmetric 3D shapes, and two other industrial cases. In all the considered cases, the design methodology achieved substantial reductions in the forging load without forging defects, proving its reliability and effectiveness for application in metal forging processes.

**Keywords:** forging process; preform design; convolution neural network (CNN); CNN sub-models

## 1. Introduction

The utilization of the preforming step is important in metal forging processes because it avoids the manifestation of forging defects [1,2], reduces the forging load [3,4], and extends the die life [5,6]. However, preform design is conventionally performed by trial-and-error approaches based on an engineer's experience and know-how [7]. Thus, systematic preform design methods have been studied in recent decades.

By synthesizing the design rules in the reported literature and the experts' know-how, an empirical equation for H-shape forging [8], a knowledge-based system for rib-web type forgings [9], and an expert system for gear forgings [10] were developed to suggest the preform shapes. Although these approaches are sufficiently good for certain simple forgings, they are not practical because the guidelines are insufficient for complex geometries.

To overcome this limitation, Park et al. [11] introduced the backward tracing scheme (BTS) method to determine a proper preform design by using the H-shapes and their boundary conditions and inversely tracing the loading path. In two-dimensional (2D) axisymmetric forging cases, Kim and Kobayashi [12] applied the BTS method to closed-die forging, and obtained good results in terms of the metal flow for the predicted preform. Gao et al. [13] extended this methodology to the preform design of three-dimensional (3D) blade forging by considering a non-uniform deformation. However, the BTS methodology requires a user's initial guess for the preform design, which means that the result of this

method is user-biased. Moreover, a designed preform is also influenced by the number of iterations and the loading path, which are not included in the design requirements.

Badrinarayanan and Zabaras [14] introduced a linear sensitivity analysis (LSA)-based preform design method for axisymmetric forging. LSA approaches consider the change in the node position and compute the finite-dimensional gradients of the objective functions. Zhao et al. [15] designed a preform for H-shaped forging considering the sensitivities of the objective functions and nodal velocities. However, because the governing equations in LSA approaches are strongly related to the forging shape, they become overcomplicated in the case of complex geometries. More recently, Lu et al. [16] applied evolutionary structural optimization (ESO) by considering progressive element elimination during the procedure and applied it to the preform design of H-shaped geometry. Shao et al. [17] and Yang et al. [18] considered a bidirectional ESO method in which the elements eliminated in a certain step could be recovered in subsequent steps. However, this ESO method also requires an initial guess for the preform geometry, which becomes user-biased and is affected by aleatoric uncertainty.

To suggest a preform design shape without any initial design guess, Roy et al. [19] utilized an artificial neural network (ANN) algorithm for H-shape forging. The ANN-based model utilizes the weight vectors and links the values of the forging geometry image to those of the preform image in a one-to-one relationship. Pathak et al. [20] also applied the ANN method to a cylindrical billet upsetting process to reduce the barreling at the free surface of the workpiece. However, as the ANN method assigns the weight considering a one-to-one relationship for the user-defined training data, this method yields a poor level of detectable details and becomes impracticable for untrained complex forging shapes.

In contrast, a convolutional neural network (CNN) algorithm can extract the geometrical features of input data arrays through convolutional operations with weight arrays (filters) [21]. As this algorithm does not require a one-to-one link between data, it has been widely employed in several engineering problems [22,23]. After the convolution operation, an array is defined as a feature map, where the extracted features of the input data are stored. The extracted feature maps are connected to an output data array by multiplying them by weights. All the utilized filters can be considered as a cumulative experience for engineers to infer appropriate output data from the input data.

Therefore, a new preform design approach based on the CNN algorithm is introduced. As the implemented CNN algorithm imitates the preform design process of the forging experts, the trained filters are expected to act as a sort of cumulative experience, utilized for the prediction of the preform shapes. Through the convolution operation with the filters, the feature maps relevant to the inputted forging shapes are identified, and the appropriate preform design is automatically determined by applying them to transform the forging geometry. Because the design of a 3D preform shape is normally a complicated and trial-and-error task, it is intended to be able to obtain multiple 3D preform candidates based on the capability of trained CNN sub-models for estimating 3D geometries. During the training procedure, filters were constructed based on the characteristics of the specified design requirements. Accordingly, we intended to derive a preform satisfying the design criterion considered in the corresponding training database. Even if different design criteria are considered during the construction of the training database, the proposed procedure can still be directly applied following the logic presented in this paper.

## 2. Preform Design Methodology

A flowchart of the CNN-based preform design model is presented in Figure 1. The training database consisted of forging products and relevant preform shapes. The training materials should be designed to satisfy design requirements without user bias to eliminate the limitations that appeared in the literature survey. The product geometry was converted into a 3D digital data array (voxel array), and the preform candidates were derived from the various trained CNN models. The derived 3D data arrays of the preform candidates were converted into 3D modeling geometries. To select the best preform among the candidates,

each candidate was evaluated by a finite element method (FEM) simulation. Based on the FEM results, one shape was easily selected as the preform that satisfied the design requirements of the input forging product.

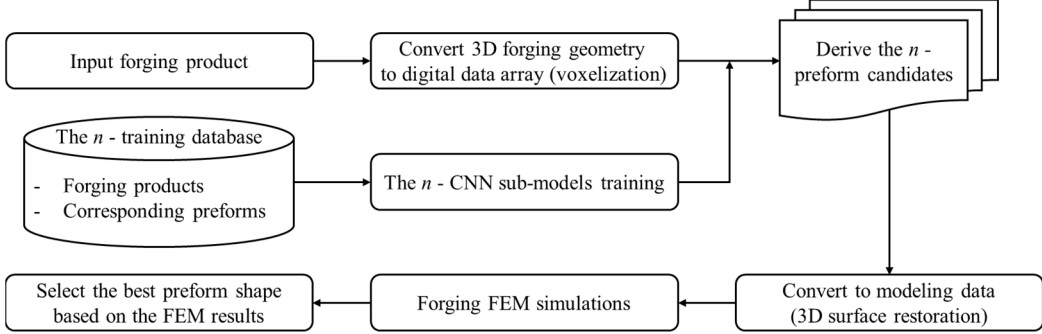

**Figure 1.** Flowchart for the preform design methodology.

## 2.1. Convolution Neural Network Model Structure

The CNN model utilized in this paper is based on the U-shape architecture [24]. The model is composed of nine convolution blocks with two convolutional layers and eight pooling layers, as shown in Figure 2. For the application of 3D geometries, the convolution operation is conducted in 3D digital volume space (voxel array).

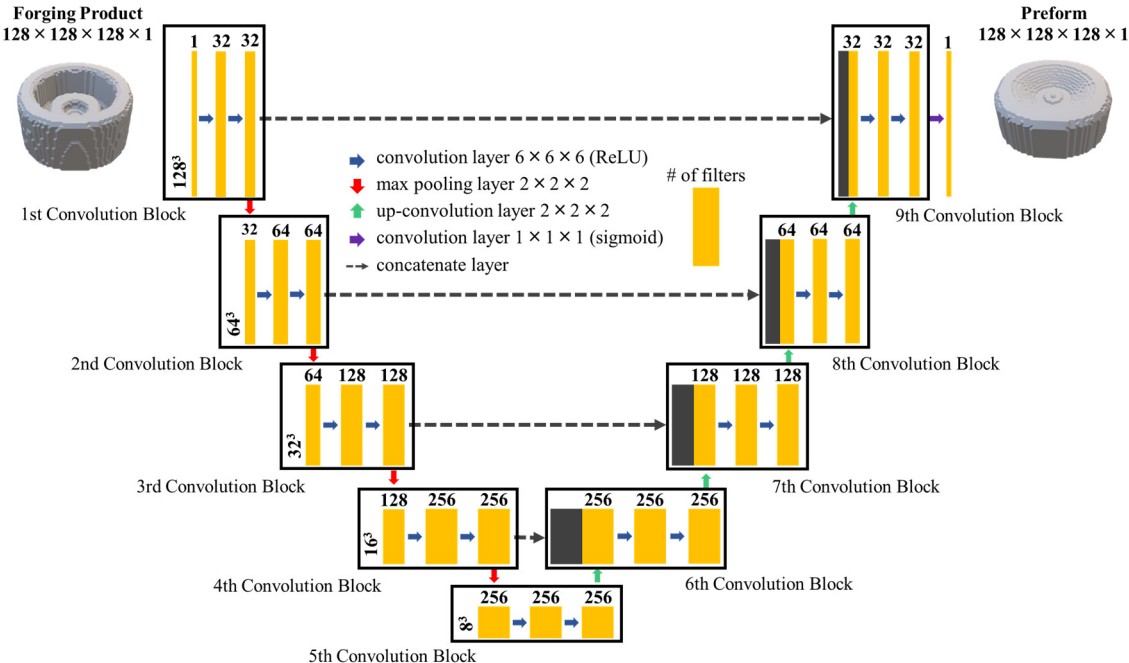

**Figure 2.** CNN model overall structure.

The voxel array input to each layer is ($h \cdot w \cdot d \cdot c$). $h$, $w$, and $d$ are the height, width and depth of the array, respectively, and $c$ is the number of channels that signifies the information of the array. The convolution operation is conducted as the filters move in the $h$, $w$, and $d$ directions of the input array. In this study, 128 voxels were considered for the $h$, $w$, and $d$ values in the voxel array of the 3D forging and preform geometries through the conversion processes, summarized in Appendix A. For the filter array, six voxels for each of the three axes were considered. Those values are the trade-off for the training convergence and computational efficiency.

The feature recognition process for the $i$-th convolution layer begins from the input data array $n \times n \times n \times c_{i-1}$ ($c_{i-1}$ is the number of the filters of the previous layers). To avoid

dimension loss during the convolution operation, zero values are padded to the outside of an input array, resulting in $(n + 2) \times (n + 2) \times (n + 2) \times c_{i-1}$. After the convolution with $c_i$ filter arrays of $6 \times 6 \times 6 \times c_{i-1}$, the output array is $n \times n \times n \times c_i$ (feature maps). The procedure in the first convolution layer is illustrated in Figure 3 where $n$ is 128, $c_i$ is 32 and $c_{i-1}$ is 1.

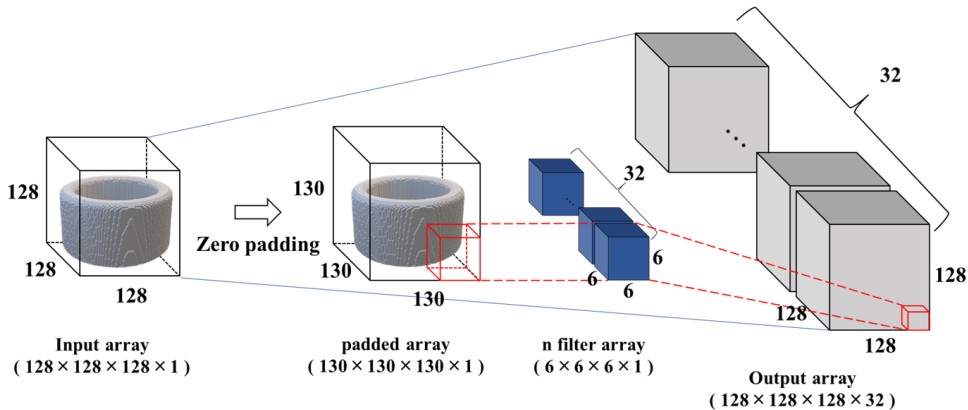

**Figure 3.** Procedure in the first convolutional layer.

The $n \cdot n \cdot n \cdot c_i$ array is reduced in half the size of $n/2 \cdot n/2 \cdot n/2 \cdot c_i$ by the red arrows in Figure 2 which represent a max-pooling operation. After that, by doubling the number of filter arrays, the features recognized in the previous convolution block are "zoomed in". After the 4th convolution block, the size of the considered image is increased to derive the preform shape in the original dimensional scale. In this upgoing sampling procedure following the green arrows, the feature maps are mapped to the considered preform design, starting from small features (convolution block 5) to large features (convolution block 9). By concatenating the number of features in the down-sampling and the up-sampling operation, any loss of information concerning the positions of the feature positions in both the forging and preform shapes is avoided.

The general formulation of the convolution operation is defined in Equation (1). $x$ is the inputted array, $w$ is the filter array, and the bias $b$ is added in computation. $p$, $q$, and $r$ are the coordinates in the feature maps. The predicted value is calculated as the output of the activation function $f$. The weights in filters ($w$) are determined to minimize the loss between the true value $y$ of the given preform shapes and of the CNN model. Since the inputted data have binary values, a binary cross-entropy loss function $E$, defined in Equation (2), is utilized. For the minimization process, an ADAM (Adaptive Moment Estimation) optimizer [25] as defined in Equation (3) is utilized assuring the robust and quick weight update process by a two-moment vector, $\beta_1$ and $\beta_2$. For the activation function, a ReLU (Rectified Linear Unit) function in Equation (4) is utilized for all the convolution layers except for the last one. Since the output of the ReLU function is $x_i$ in the $x_i > 0$ domain, the $dE/dw_i$ in Equation (3) cannot become zero. (If it becomes zero, the relevant weight is no longer updatable.) Since the last layer is utilized to predict the binary values of the array for a predicted preform shape, the sigmoid function in Equation (5), whose range is $0 \leq f(x) \leq 1$, has been applied.

The CNN algorithm is implemented in Python 3.7.4 (Python Software Foundation, Wilmington, USA) and Keras 2.3.1(Keras, Mountain View, USA) with the Tensorflow backend framework on an NVIDIA V100-SXM2 GPU and 180 GB RAM computer systems. The hyperparameters utilized in the training procedure were determined through the random search method consisting of 300 training steps, and are equal to $\varepsilon = 10^{-8}$, $\beta_1 = 0.9$, $\beta_2 = 0.999$, and $\eta = 5 \cdot 10^{-5}$.

$$\hat{y}_{pqr} = f\left(\sum_m \sum_{i=1,j=1,k=1}^{6} x_{(p+i)(q+j)(r+k),m} \cdot w_{ijk,m} + b\right) \tag{1}$$

$$E(y_i, \hat{y}_i) = -\frac{1}{n} \sum_{i=1}^{n} y_i \log(\hat{y}_i) + (1 - y_i) \log(1 - \hat{y}_i) \quad (2)$$

$$w_i = w_{i-1} - \eta \frac{\hat{m}_{1,i}}{\sqrt{\hat{m}_{2,i} + \varepsilon}}$$

$$where\ m_{n,i} = \beta_n m_{n,i-1} + (1 - \beta_n) \left( \frac{dE}{dw_{i-1}} \right)^n and\ \hat{m}_{n,i} = m_{n,i-1}/(1 - \beta_n^i),\ n = 1, 2 \quad (3)$$

$$f(x) = \begin{cases} 0\ for\ x \leq 0 \\ x\ for\ x > 0 \end{cases} \quad (4)$$

$$f(x) = \frac{1}{1 + e^{-x}} \quad (5)$$

Since the derived array of preform shapes had a range from 0 to 1, a threshold of 0.7 has been employed, meaning that if the output of the sigmoid activation function is greater or equal to 0.7 the relevant position is considered as part of the preform, whereas the position is neglected. The overall CNN-based preform design concept is summarized in Figure 4. To enhance the understanding of the proposed methodology, the explanation reported in Figure 4 is based on 2D images of the forging product and preform shapes.

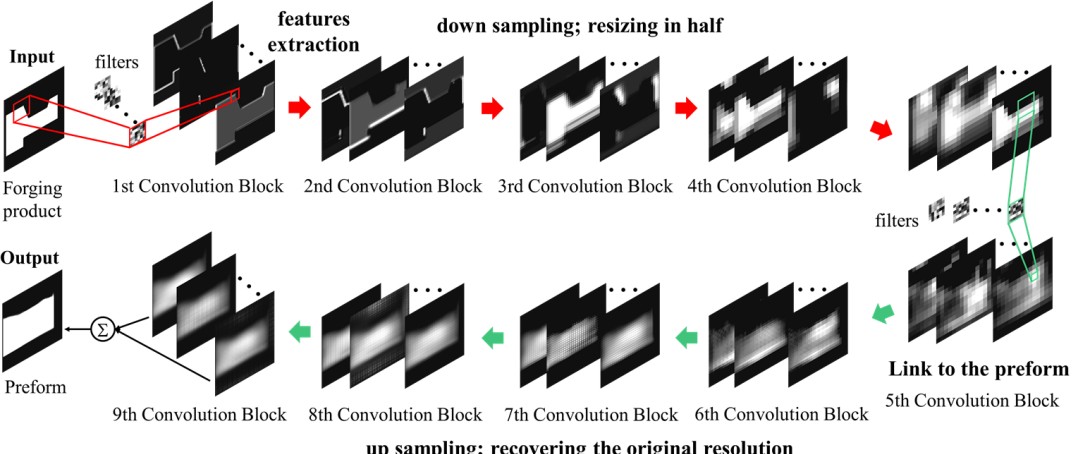

**Figure 4.** The preform design step during the implementation of the CNN algorithm.

### 2.2. Training Database Definition

A training database consisting of the forging shapes and the corresponding preform shapes should be constructed to avoid user bias. According to the preform shape, in the training database, satisfying the objective design criterion, the weight values of the filters can be set to meet the considered criterion. In this study, both the forging and preform shapes for the training database were parametrically designed to avoid any bias. The design criterion was defined as the reduction in the forging load and the avoidance of metal flow overlapping and die unfilling.

Regarding the forging shape, a combination of four levels (0.24, 0.36, 0.48, and 0.6 D) at four heights (H1, H2, H3, and H4) was utilized to design all 96 geometries with the same outer diameter D, as shown in Figure 5a. All cases had similar volumes in the $-2\% \leq V \leq 2\%$ range. An R = 5 mm filet radius was used to connect the four regions for all cases. An example shape is shown in Figure 5b.

To define the preform shape geometries, three levels (0.3, 0.4, and 0.5 D) were considered for five heights from H1 to H5. A B-spline function was utilized to generate 240 preform geometries, as shown in Figure 6a. H6 was changed to match the volume of the forging shapes. Two examples of preform shapes are shown in Figure 6b. The parametric design procedure was implemented in SolidWorks 2016 (Dassault Systèmes SolidWorks Corporation, Waltham, MA, USA) CAD software. To reduce the computation

time, the forging shapes were considered as axisymmetric with a flat surface, and the preform shapes were presented using B-spline in axisymmetric shapes.

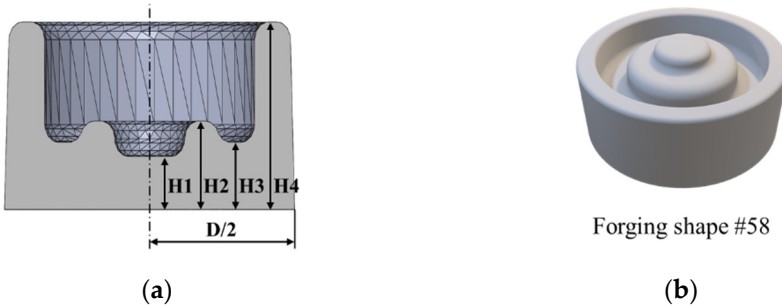

| (a) | (b) |

**Figure 5.** (**a**) Schematic diagram of the geometrical features of the forging shapes, (**b**) forging shape #58 (example).

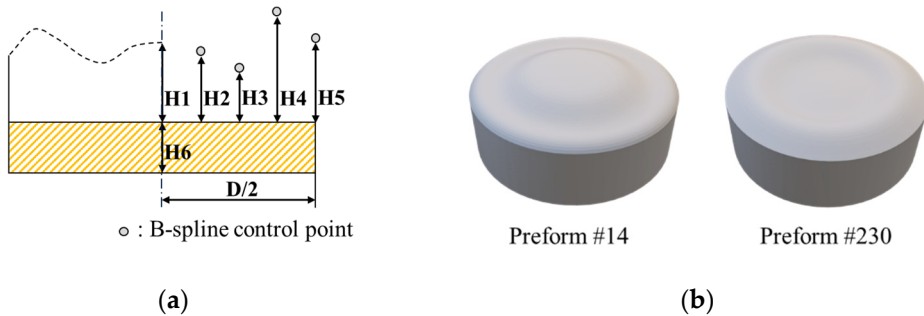

| (a) | (b) |

**Figure 6.** (**a**) B-spline method for the preform shapes, and (**b**) preform #14 and #230 (examples).

Among the 240 preform shapes, one shape satisfying the defined design rules was selected as the preform shape of the corresponding forging shape based on the FEM simulation results. The FEM simulations were implemented in the DEFORM 2D Version 11.0 (Scientific Forming Technologies Corporation, Columbus, OH, USA) software, as summarized in Appendix B. The preform billet is defined as a deformable plastic considering AA6061 aluminum alloy using the Ludwick-type flow stress in Equation (6), calibrated by tensile test experiments on ASTM-E8 specimens at room temperature.

$$\bar{\sigma} = 225.43\bar{\varepsilon}^{0.257} + 144.79 \text{ [MPa]} \tag{6}$$

Although the AA6061 aluminum alloy was utilized in the implemented FEM model, the proposed procedure is independent of the material, as will be shown in the results section. Because the objective of the FEM investigation is to compare the forging load reduction and the plastic flow of the preform, and not a detailed investigation of the stress distribution, all simulations were implemented under cold and isothermal forging conditions.

*2.3. Sub-Models Definition*

By defining the multiple CNN sub-models, the filters have different values for each sub-model. Thus, individual preform candidates can be derived according to each CNN model. This subdivision strategy allows the final shapes that share similar geometrical features to be grouped together, increasing the level of accuracy of the relevant CNN sub-models.

In this study, six data groups were sorted by considering the geometrical characteristics (H1, H2, H3, and H4) of the forging shape, as shown in Figure 7.

| CNN sub-model # | CNN#1 | CNN#2 | CNN#3 | CNN#4 | CNN#5 | CNN#6 |
|---|---|---|---|---|---|---|
| Height-based grouping conditions | H1 < H3 < H2 | H3 < H1 < H2 | H2 < H1 < H3 | H2 < H3 < H1 | H3 < H2 < H1 | H1 < H2 < H3 |
| | H2 < H4 < H3 | H4 < H2 < H3 | H3 < H2 < H4 | H3 < H4 < H2 | H4 < H3 < H2 | H2 < H3 < H4 |
| Level # | 32 | 32 | 32 | 32 | 28 | 28 |

**Figure 7.** Geometry-based grouping procedure for the six CNN sub-models with shape examples.

According to the defined grouping criterion in Figure 7, one forging shape may belong to one single group or two different groups, resulting in each group having either 28 or 32 levels. The subdivision of the database was utilized for six CNN sub-models, all with the same CNN architecture. Six preform candidates were derived for an inputted forging shape according to each sub-model and further evaluated through FEM simulations to select the best preform that reduces the forging loads and avoids forging defects.

## 3. Numerical Experiments

For each of the multiple CNN sub-models, a range of forging load reductions could be achieved according to the corresponding training dataset, in comparison with a cylindrical billet with the same volume of the forging shape. The minimum and maximum forging load reductions for the data groups utilized for training the six CNN sub-models are summarized in Table 1.

**Table 1.** Average forging loads reduction in six CNN sub-models.

| CNN Sub-Model # | CNN#1 | CNN#2 | CNN#3 | CNN#4 | CNN#5 | CNN#6 |
|---|---|---|---|---|---|---|
| Range of Forging Loads Reduction | 11.63–26.53% | 10.32–27.89% | 11.63–21.89% | 9.64–26.53% | 11.86–25.42% | 9.64–27.89% |

The proposed preform design model was applied to two 2D axis-symmetric forging cases in Section 3.1, $\frac{1}{4}$ plane-symmetric forging in Section 3.2, and grinding tool-like forging in Section 3.3.

### 3.1. Two Axisymmetric Forging Cases

The CNN sub-models were utilized to derive preform shapes for the two axisymmetric test parts, as shown in Figure 8 where the height features are listed in Table 2. The dimensions of the two shapes were randomly designed to yield completely different geometrical features.

The finite element forging simulation results for the preform candidates were acquired from the 2D FEM model, summarized in Appendix B, by considering the 2D cross-sections of the preform candidates as derived by 3D modeling shapes.

Based on the FEM results, for the test shape, as shown in Figure 8a, the preform derived by the CNN#3 sub-model resulted in a maximum forging load reduction of 15.9% (Figure 9). For the shape shown in Figure 8b, the CNN#4 sub-model derived a preform resulting in a 16.5% forging load reduction (Figure 10).

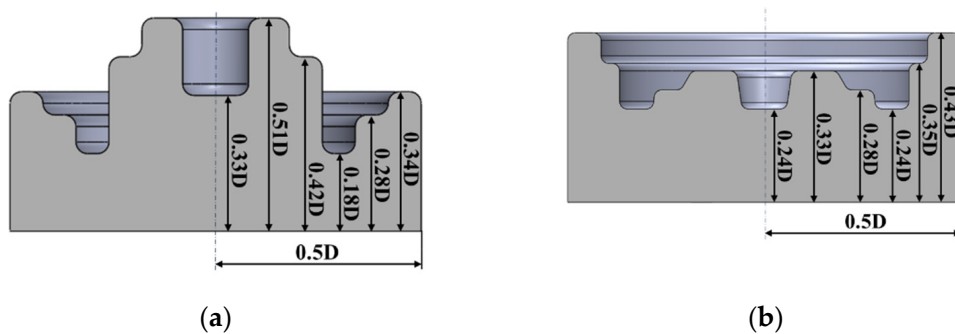

**Figure 8.** (**a**) First and (**b**) second axisymmetric test forging shapes adopted for the validation of the CNN sub-models.

**Table 2.** Height features of the derived preform shapes for the axisymmetric forging shapes.

| Forging Shape | Height | CNN#1 | CNN#2 | CNN#3 | CNN#4 | CNN#5 | CNN#6 |
|---|---|---|---|---|---|---|---|
| Figure 8a | $H_{max}$ | 0.48D | 0.51D | 0.51D | 0.50D | 0.50D | 0.48D |
| | $H_{min}$ | 0.30D | 0.29D | 0.32D | 0.30D | 0.30D | 0.29D |
| Figure 8b | $H_{max}$ | 0.40D | 0.39D | 0.40D | 0.40D | 0.39D | 0.41D |
| | $H_{min}$ | 0.27D | 0.29D | 0.31D | 0.28D | 0.33D | 0.30D |

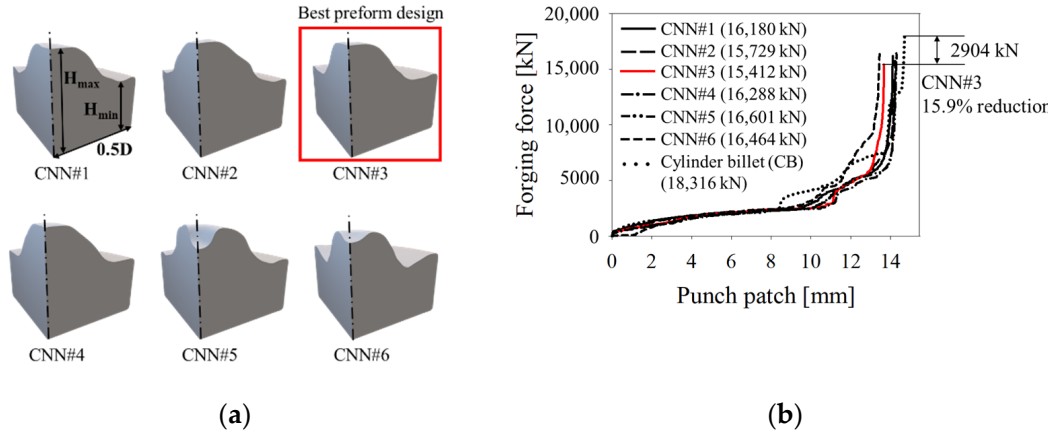

**Figure 9.** (**a**) Preform geometries and (**b**) forging force–punch patch curves computed for each of the CNN sub-models (including the cylindrical billet) for the forgings of Figure 8a.

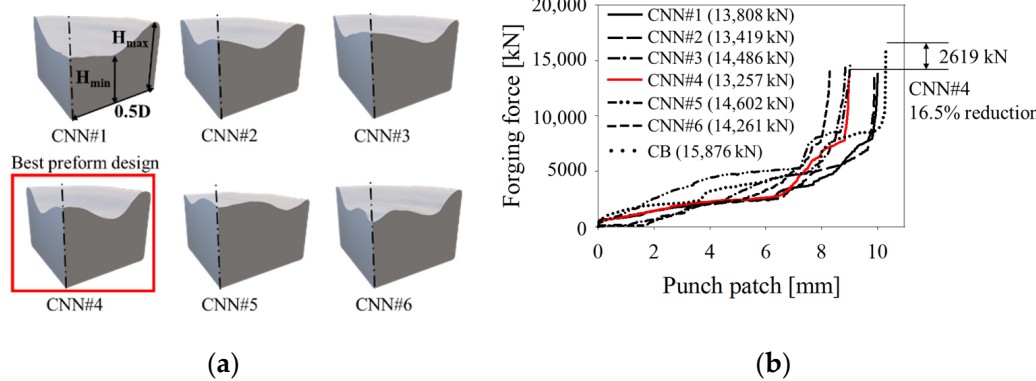

**Figure 10.** (**a**) Preform geometries and (**b**) forging force–punch patch curves computed for each of the CNN sub-models (including the cylindrical billet) for the forgings of Figure 8b.

### 3.2. The One-Quarter Plane-Symmetric Forging

The CNN models were trained by utilizing 3D data arrays; thus, the trained models could derive the 3D preform shapes for the 3D inputted forging geometries. However, the training dataset used in this study was defined by considering axis-symmetric forgings. Therefore, to investigate the capability of the CNN sub-models trained with axisymmetric 2D images, and to accurately derive the preform shape for the case of non-axisymmetric shapes, two strategies were implemented for a $\frac{1}{4}$ plane-symmetric simple forging, as shown in Figure 11. The first consisted of directly inputting the 3D images into the CNN sub-models. The second consisted of inputting separately the axisymmetric shapes of three cross-sections along $0°$, $45°$, and $90°$.

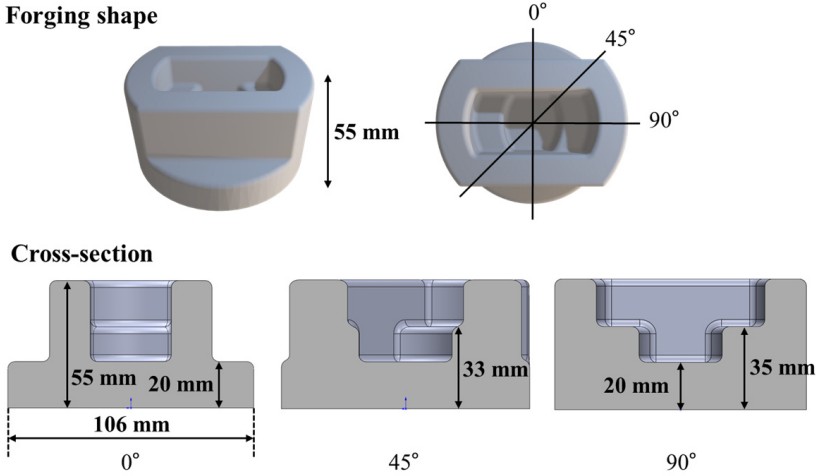

**Figure 11.** The 1/4 plane-symmetric forging with cross-sections along $0°$, $45°$, and $90°$ directions.

Considering the first approach, the six shapes of the preform candidates are shown in Figure 12a and Table 3 and the preform shape derived by CNN#2 ensured the highest forging load reduction of 11.1% compared with that of a cylindrical billet (Figure 12b). The FEM results presented in this section were acquired through a 3D thermo-mechanical FEM model, as described in Appendix B.

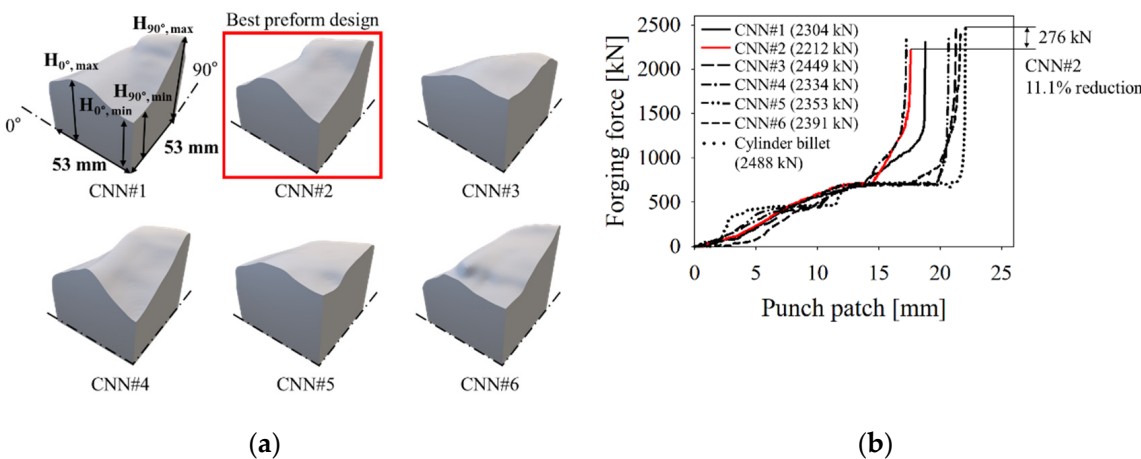

**Figure 12.** (**a**) Preform geometries for the symmetric forging and (**b**) forging force–punch patch comparison.

**Table 3.** Height features of the derived preform shapes for the symmetric forging.

| Height [mm] | CNN#1 | CNN#2 | CNN#3 | CNN#4 | CNN#5 | CNN#6 |
|---|---|---|---|---|---|---|
| $H_{0°, max}$ | 39.44 | 43.05 | 42.41 | 41.64 | 42.83 | 38.11 |
| $H_{0°, min}$ | 30.43 | 30.11 | 31.36 | 28.39 | 32.55 | 31.79 |
| $H_{90°, max}$ | 54.64 | 51.19 | 43.54 | 52.67 | 43.54 | 55.28 |
| $H_{90°, min}$ | 30.43 | 30.11 | 35.21 | 28.39 | 36.01 | 33.64 |

Regarding the second approach, the three preforms estimated for each of the three considered directions ($0°$, $45°$, and $90°$) were combined into a single image by a Boolean union and smoothed in the transition regions between the three angles.

Considering the results reported in Figure 12b, the preform shape derived by CNN#2 resulted in a maximum forging load reduction. For this reason, the preform shapes by inputting the 3D arrays to CNN#2 and by inputting the cross-sections at $0°$, $45°$, and $90°$ to CNN#2 were utilized in two forging simulations, and the relevant forging force versus stroke results are shown in Figure 13b.

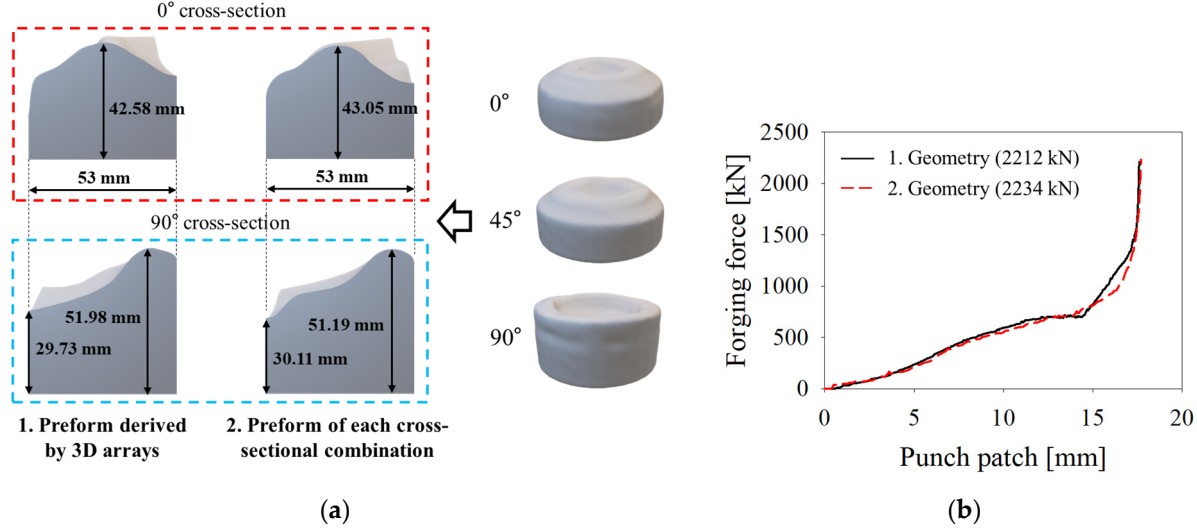

(**a**)  (**b**)

**Figure 13.** (**a**) Preform shapes derived by CNN#2 for a 3D model inputted at once and the case of combining the $0°$, $45°$, and $90°$ shapes and (**b**) Forging force vs. punch patch comparison for two preforms.

The comparison between the $0°$ and $90°$ cross-section dimensions of the two preforms derived through these two methods shows that they are remarkably similar in both the overall shape and dimensions. This indicates that although the training dataset was defined by considering axis-symmetric forgings, the proposed method can be applied to forging geometries in which at least one symmetry plane can be identified.

### 3.3. Grinding Tool-Like Forging

To test the reliability of the proposed approach, a literature case relevant to grinding tool-like forging [26] was considered, as shown in Figure 14. In the considered geometry, two symmetry planes were identified, as shown in Figure 14.

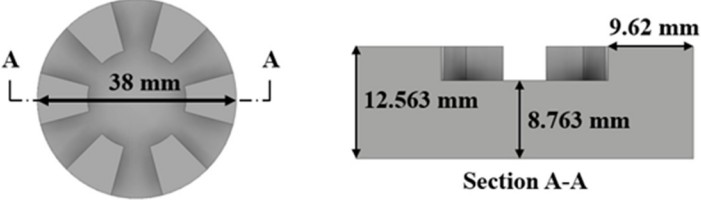

**Figure 14.** The grinding tool-like forging shape [26].

The one-quarter models of the derived preform shapes are shown in Figure 15 and Table 4. For the estimation of the forging load reduction, 3D plane-symmetric FEM simulations were implemented in DEFORM 3D, as summarized in Appendix B, considering the plasticine material reported in [27].

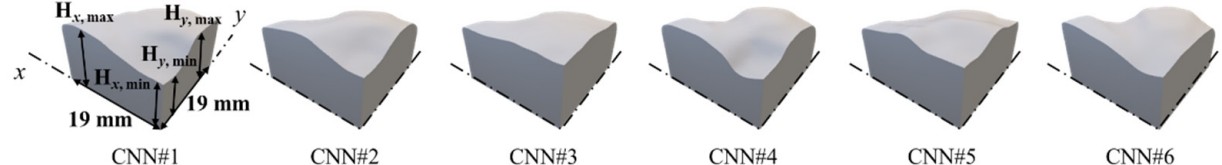

**Figure 15.** Preform shapes by CNN sub-models for the grinding tool-like shape.

**Table 4.** Height features of the derived preform shapes for the grinding tool-like shape.

| Height [mm] | CNN#1 | CNN#2 | CNN#3 | CNN#4 | CNN#5 | CNN#6 |
|---|---|---|---|---|---|---|
| $H_{x,\,max}$ | 11.40 | 11.20 | 10.90 | 11.10 | 11.30 | 11.50 |
| $H_{x,\,min}$ | 7.94 | 8.77 | 10.48 | 7.63 | 9.98 | 8.50 |
| $H_{y,\,max}$ | 10.80 | 10.60 | 11.20 | 10.80 | 11.10 | 9.84 |
| $H_{y,\,min}$ | 7.37 | 6.51 | 7.44 | 6.04 | 7.80 | 9.13 |

As shown in Figure 16, all six preform shapes allowed a maximum forging load reduction in comparison with the cylindrical preform, and the maximum reduction was identified for the preform shape derived by the CNN#5 sub-model. In Figure 16c, the comparison between the experimental results [26] and the authors' FEM model results for the cylindrical billet show a particularly good agreement, proving the reliability of the implemented FEM model. However, the preform design derived by the proposed methodology reduced the forging load and eliminated forging defects.

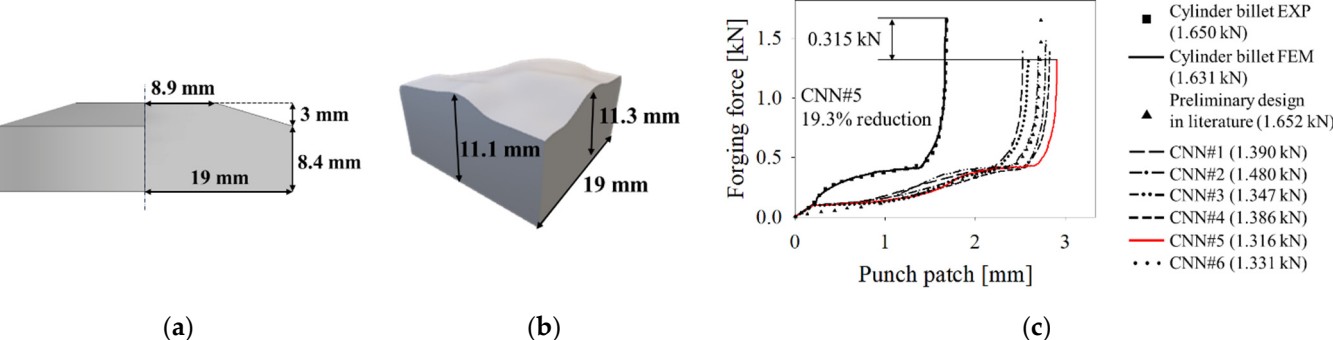

(**a**)         (**b**)         (**c**)

**Figure 16.** (**a**) The preform design from the literature [26], (**b**) preform design by CNN#5, and (**c**) experimental forging force–punch patch curves from the literature from FEM for the designed preforms by CNN sub-models.

## 4. Experimental Verification

To verify the applicability of the proposed preform design approach to a real industrial case, piston head forging (Figure 17) was considered. This forging component is a one-quarter plane-symmetric shape characterized by thin walls and is designed for weight reduction.

The piston head geometry was input to the trained CNN sub-models and the derived preform shapes are shown in Figure 18 and Table 5. The derived preform shapes were utilized in the 3D thermo-mechanical FEM simulations, as described in Appendix B. As for the industrial manufacturing process, the preforming stage can prevent die failure by reducing the die wear depth, which is significantly related to the forging load. Therefore, for the piston head verification step, the die wear depth according to the preform shapes was also estimated by the FEM investigations. Regarding the preform shapes derived

by CNNs #4, #5, and #6, some features were considered too complicated to be obtained during a preform-forging phase, resulting in folding defects during the piston-forging phase, as highlighted in Figure 18. Therefore, these three candidates were excluded from further investigation.

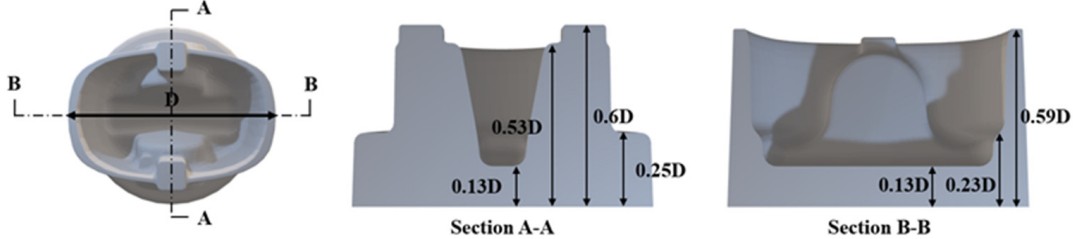

**Figure 17.** Piston head forging and dimensions.

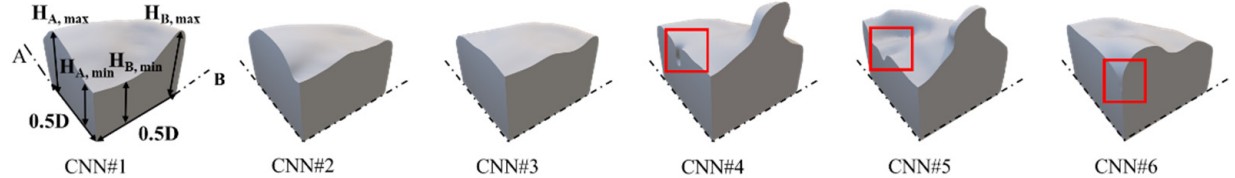

**Figure 18.** Preform shapes for the piston head forging with excluded preform geometries.

**Table 5.** Height features of the derived preform shapes for the grinding tool-like shape.

| Height [mm] | CNN#1 | CNN#2 | CNN#3 |
|---|---|---|---|
| $H_{A, max}$ | 0.48D | 0.51D | 0.51D |
| $H_{A, min}$ | 0.30D | 0.29D | 0.32D |
| $H_{B, max}$ | 0.38D | 0.32D | 0.34D |
| $H_{B, min}$ | 0.24D | 0.29D | 0.31D |

As shown in Figure 19, the remaining three CNN-based preforms enabled the complete filling of the die, and the preform based on the CNN#3 model yielded the largest forging force reduction, estimated to be 15.09%, and promoted a wear depth reduction, estimated to be 16.05%, in comparison with the cylindrical billet.

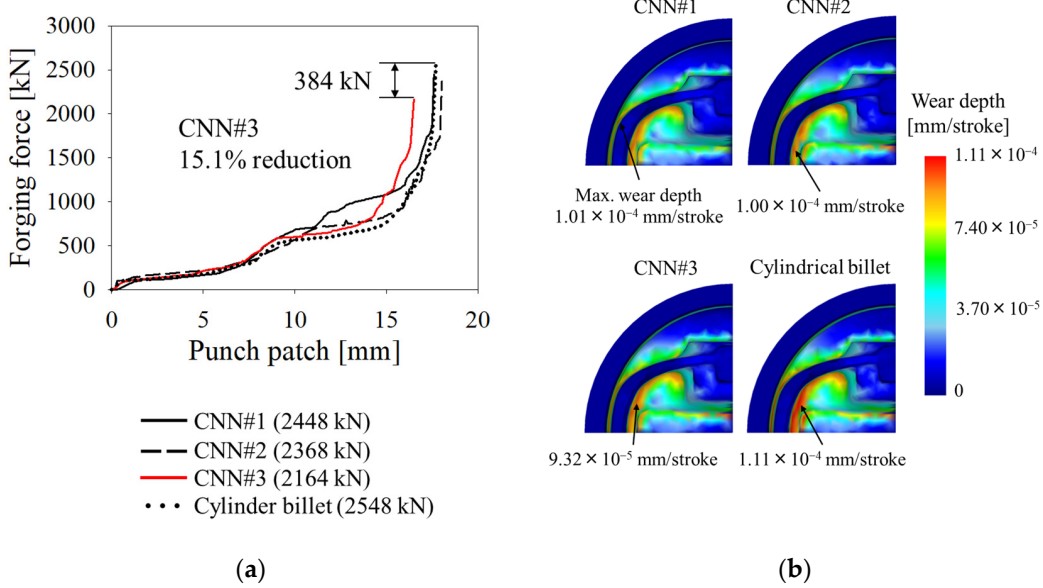

**Figure 19.** (**a**) Forging force–punch patch curves and (**b**) die wear distribution for CNN#1, CNN#2, CNN#3, and cylindrical billet preforms, respectively.

Based on the positive results obtained from the FEM investigation in Figure 19a, two experiments were performed considering the CNN#3 preform design and the cylindrical billet. For the hot forging process, the HKLP-800 press machine shown in Figure 20a was utilized, and the set-up conditions for the process summarized in Appendix B were utilized in the 3D FEM simulations. The top and bottom dies, relevant for the piston head shape, are shown in Figure 20b.

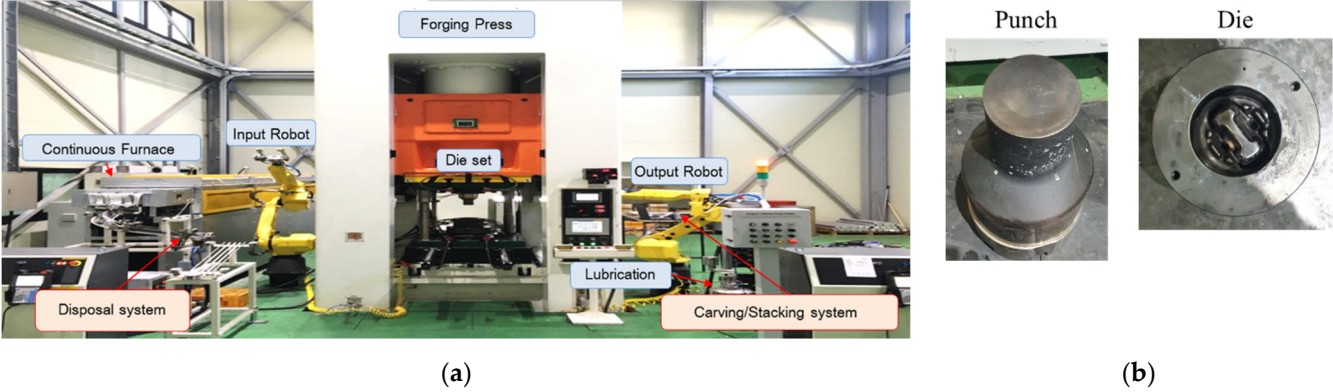

(**a**)  (**b**)

**Figure 20.** (**a**) Experimental set-up and (**b**) punch and die utilized in the verification experiments.

Because the preform die has not been manufactured yet, the preform shape derived by CNN#3 (Figure 21a) was obtained by machining from a cylindrical billet. In addition, an experiment considering a cylindrical billet was performed for comparison. When comparing the experimental and numerical forging load–stroke curves for the cylindrical and CNN#3-based preforms, as shown in Figure 21c, the deviation between the experimental and FEM results was limited to 1.05% and 0.55% for the area beneath the curves and 1.95% and 0.93% for the case of the maximum load, respectively. These limited errors indicate that the implemented 3D thermo-mechanical FEM model can properly replicate the considered forging process. In addition, the cross-section of the forged piston head (Figure 21b), cut through computer numerical control machining, shows the high quality of the filling as well as the absence of any overlap of the metal flow. This result proves the reliability of the defined CNN-based preform design approach and the accuracy of the implemented 3D thermo-mechanical FEM model in replicating the considered forging process conditions.

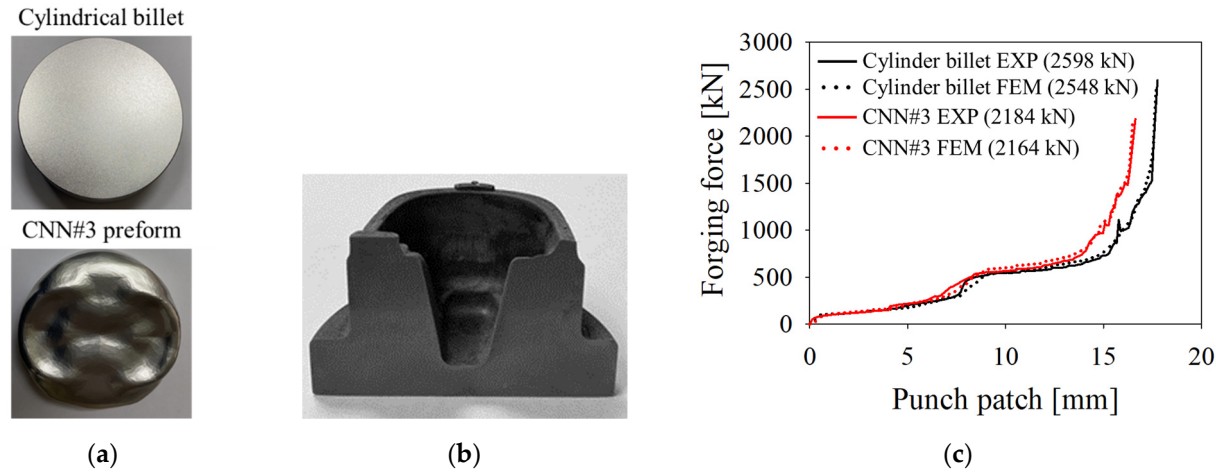

(**a**)  (**b**)  (**c**)

**Figure 21.** (**a**) Cylindrical billet and CNN#3-based preforms, (**b**) cross-section of the forged piston head with the CNN#3-based preform, and (**c**) forging force–punch patch curves obtained from experiments and FEM simulations.

## 5. Discussion

As the creation of the training database is a time-consuming process, certain simplifications must be implemented, as performed in this study. Two-dimensional axisymmetric geometries were considered in the definition of the training database for both forging and preform shapes. For comparison, the 2D axisymmetric simulations required an average of 10 min each, whereas the 3D thermo-mechanical simulation for the piston head forging required approximately 2 h. For this reason, the ability to create a simplified, but still reasonable, training database is important to achieve accurate results without consuming excessive computational time. Moreover, the consideration of a cold-forging FEM can also reduce the computational time. Although the material flow resistance is strongly influenced by the forging temperature, this simplification does not alter the accuracy of the preform design because the material flow itself is independent of the processing temperature.

As presented in Sections 3 and 4, the CNN sub-models automatically derived the 3D surface of the preform candidates by utilizing the trained filters. Considering the first validation phases, relevant for the application of the trained CNN sub-models to two additional axisymmetric forgings and a $\frac{1}{4}$ plane-symmetric forging, the results of the FEM investigations demonstrated the capability of the proposed preform design approach in deriving a preform shape that allows a significant reduction in the forging load by at least 11.1%. Regarding the grinding tool-like forging, the comparison between the experimental and FEM results validated the 3D forging FEM model implementation strategy and showed an improvement in the preform shape compared to that proposed in the relevant reference study [26]. Regarding the investigation of the piston head forging described in Section 4, the utilization of the preform shape derived by CNN#3 resulted in a reduction in the forging load of 15.09% (FEM results) and 15.94% (experimental result), as well as in the wear depth reduction estimated at 16.05%. Hence, the possibility of designing individual preform shapes, regardless of the material, allows the user to choose among the derived shapes based on the maximum force criterion and considering possible constraints for the preform manufacturing process.

Although the present study was mainly focused on forging load reduction, additional target parameters can be included as target functions for the training database definition. For instance, damage or residual stress minimization can be added as a criterion for the preform shape. However, the complexity of the resulting FEM model hinders its application to complex geometries, owing to the unreasonable amount of computational time required for the definition of the training database. This issue is critical for complex problem solutions by means of machine learning algorithms and requires more work in the future.

Furthermore, the training phase in the current study was based solely on the results of numerical simulations; however, it can also be linked to engineers' experience by feeding previously validated sets of preforms and final shapes along with the FEM results. This combined approach is particularly interesting because the robustness of the training data can be further extended without requiring any additional computation time.

## 6. Conclusions

In this study, a new preform design methodology based on a CNN was proposed and investigated for a certain group of forging processes. The user-bias limitations highlighted in previously published algorithms can be overcome by considering a parametric design approach in the training database definition. The filters were successfully determined during the convolutional operations or the so-called training procedures by extracting the geometrical features of the forging product and linking them to the corresponding preform shapes, such as the building of a human design experience. Based on the multiple 3D CNN model strategies, multiple 3D preform design candidates for one inputted forging product geometry were easily acquired. Therefore, it is possible to select the best one depending on the design requirements and on the results of the forging simulation for the individual preform candidates. Although the CNN sub-models were trained three-dimensionally with 2D axisymmetric cases for simplicity, the results showed that they can be successfully

extended to 3D complex forging shapes with symmetry planes. The filters were determined according to the design rules defining the training database; thus, according to the considered training set, additional design rules can be included and considered in the design of the preform geometry. It should be mentioned that the present method is a starting point for the automatization of preform design. In conclusion, the proposed preform design methodology is useful for deriving preform designs for general forging cases in the metal forging industry. However, it is expected that further investigations should be performed by including the effect of the number of training databases, the construction of a different training database, and additional manufacturing or material-related parameters.

**Author Contributions:** Conceptualization, S.L. and L.Q.; methodology, S.L.; software, S.L.; validation, S.L., I.K. and J.S.; formal analysis, S.L. and D.P.; investigation, S.L.; resources, I.K. and J.S.; data curation, S.L.; writing—original draft preparation, S.L.; writing—review and editing, S.L. and L.Q.; visualization, S.L.; supervision, N.K.; project administration, N.K.; funding acquisition, N.K. All authors have read and agreed to the published version of the manuscript.

**Funding:** This work was supported by the World Class 300 R&D Program (grant number S2317902) funded by the Ministry of SMEs and Startups, by the National Research Foundation of Korea (NRF) grant funded by the Korea government (MSIT) (No. 2019R1F1A1060567), and by the Material Component Technology Development Program (20013060) funded by the Ministry of Trade, Industry and Energy (MOTIE, Korea).

**Institutional Review Board Statement:** Not applicable.

**Informed Consent Statement:** Not applicable.

**Data Availability Statement:** All the results are available on request to the corresponding author.

**Acknowledgments:** This research was carried out with the help of the "HPC Support" Project, supported by Ministry of Science, ICT and NIPA of Korea. This support is gratefully acknowledged.

**Conflicts of Interest:** The authors declare no conflict of interest.

## Appendix A

The CAD modelling data utilized in the FEM simulation were STL mesh files; thus, in order to carry out the matrix operations in the convolution calculation in Section 2.1, they had to be converted into voxel data array in Binvox files [28,29].

For the conversion process, the array size should be determined; thus, in this research, Binvox files were created considering 122 voxels for each of the three axes, a choice which resulted in a volume loss which was lower than 1.15% in comparison to the original mesh file. The scale factor, defined to convert the mm-based dimension into voxels, is calculated between the voxel grid-scale and the longest length in the considered shape, as shown in Figure A1. The defined scale factor is saved and utilized to fit the voxel grid-scale Binvox file into a real scale in the post-processing procedure.

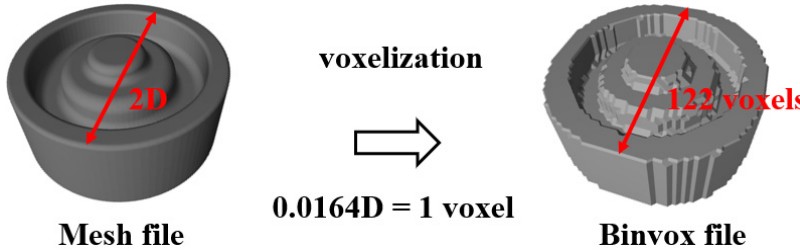

**Figure A1.** Pre-processing procedure to voxel grid-scale.

After the training process, each trained CNN sub-model predicts a preform candidate for the considered forging shape in the form of a voxel array that must be converted to a CAD file and smoothed before being utilized for the setting of the forging FEM simulation.

Firstly, the voxel array is converted to a millimeter scale of mesh shapes as a cubic frame by the Marching cube algorithm [30]. For the sake of smoothing the edge of the block-shape of the voxel, the Humphrey's smoothing algorithm [31] is applied. Since the smoothed shapes have 122 voxel diameter dimensions, the scale factor saved at the pre-processing stages is applied to match the previous dimensions and the results of post-processing as shown in Figure A2. In case of volume loss limited to less than 2%, the overall height of the preform shape is increased to match the volume of the considered forging shape.

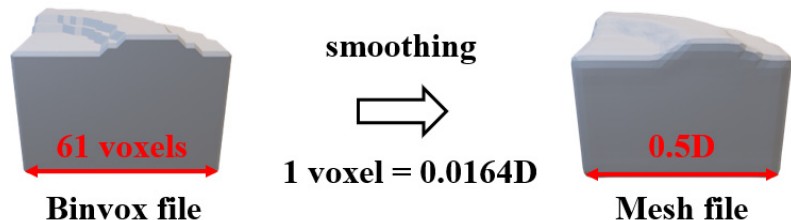

**Figure A2.** Post-processing procedure to real scale.

**Appendix B**

For the 2D axis-symmetric FEM model utilized in Sections 2.2 and 3.1, the preform billet was meshed using axisymmetric four-node elements with an element mesh density equal to 3 and average element side length equal to 0.41 mm. This parametric mesh allows a finer mesh on the corners to be obtained in contrast to the coarser mesh in the more uniform parts of the model, as shown in Figure A3a. The top die was modeled as flat and was set to move with a 1 mm/s speed.

To evaluate the preform shapes for the literature forging case in Section 3.3, a total of approximately 32,000 tetrahedral elements were utilized for the meshing of the preform. The adaptive mesh was employed resulting in the regions of the model presenting small details in having an average mesh size equal to 0.815 mm whereas the uniform regions had an average mesh size equal to 2.004 mm. Both material properties and the plastic behavior of the considered plasticine were acquired from the literature [27]. Thanks to the symmetry of the forging shape, only one-quarter of the model was considered in all the 3D simulations, as shown in Figure A3b. For the above two FEM simulation models, the top and bottom dies were considered as rigid to focus on the metal flow of the preform shape. The friction coefficient was set to 0.2.

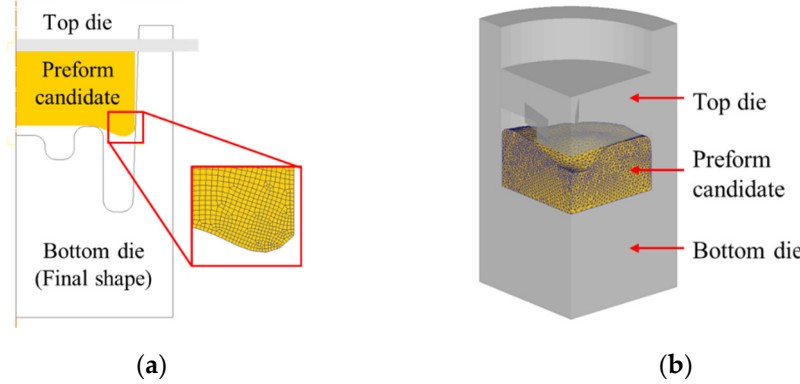

(**a**)                                (**b**)

**Figure A3.** The numerical simulation model for (**a**) 2D axis-symmetric forging and (**b**) 3D forging considering $\frac{1}{4}$ of the geometry.

For the definition of the FEM model for the piston head forging, a 3D thermo-mechanical FEM model was implemented in DEFORM 3D by considering the same mesh size utilized in the grinding tool-like shape model. The initial temperature of the preform, the die temperature, and the friction coefficient were equal to 350 °C, 280 °C, and 0.3

were those utilized in the real piston production. The AISI-H13 steel and the AA-6061 aluminum alloy were utilized for the modeling of the top and bottom dies and preform, respectively. The temperature-dependent properties and plastic behavior for both materials were acquired from the DEFORM material library and MATILDA$^{®}$ (Material Information Link and Database Service), which is largely employed for the material properties acquisition utilized in hot forging processes [32]. To determine the influence of the preform geometries on the die wear, the Archard wear model in Equation (A1) was included in the FEM simulation. The wear coefficients were calculated as $1.44 \times 10^{-6}$ for $K$, 1.63 for $a$, 1.25 for $b$, and 1.76 for $c$ based on the lab-scale pin-on-disc experimental results.

$$W = \int K \frac{P^a V^b}{H^c} dt \tag{A1}$$

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
