# Peer review of "A New Approach to Preform Design in Metal Forging Processes Based on the Convolution Neural Network"

_applsci, doi:10.3390/app11177948_

Round 1
Reviewer 1 Report
Minor comments
- Lines 288 and 316: Numbering of tables must be revised from 3 to 4 and 5.
Also, I recommend the citation of tables 2, 3, 4 and 5 in the text of the corresponding section.
- Lines 397-398 : “ …the filters could replace the forging experts’ know-how in the preform design process”. And, lines 15-16 : “the training process, emulating the accumulated experience and know-how of a preform design expert”. In my opinion, the authors could have provided sufficient precision on the choice of filters (weights) based on the experience of experts in the field.
Author Response
The authors would like to thank to reviewers' valuable comments.
We tried to address all the points thoroughly.
Please see the attachment.

Reviewer 2 Report
The manuscript has a poor scientific standard. The work presents the use of convolution neural networks in the design and optimization of forging preforms. All this to optimize the shape of the preforms and reduce the forging force. The results of theoretical research in which the program was taught with convolution neural networks were verified by simulations in the numerical program Deform. The FEM computational studies were verified by an experimental experiment in the forging laboratory. Cylinder forgings were made. The forging force reduction was investigated in numerical and experimental studies. All tested reductions corresponded to the types of learning by CNN convolution neural networks.
In numerical tests, the forging force was over 1500 tons, and in experimental tests on a press - 250 tons. Cold forging of such forgings may not be possible.
Numerical and experimental tests should be verifiable and should take place at the same temperatures.
The authors did not take into account the temperature of the cold forging ball in the numerical tests. In experimental tests, however, hot-forging bullets. This is a big inconsistency. The CNN study did not mention the material base which is necessary for teaching artificial neural networks.
Questions:
Please explain why FEM testing was performed with cold forging and hot forging?
At what temperature was forging in a laboratory experiment?
Why is there nothing material on CNN?
Amendments:
Please do not use (LOAD [tons]) only (FORGING FORCE [KN]). Please change it in the charts.
Please do not use (STROKE [mm) only (PUNCH PATCH [mm]). Please change it in the charts.
Author Response

(The authors gave the same response as above.)

Round 2
Reviewer 2 Report
Dear autors I would like to thank the authors for their explanations and honest response to comments and questions. Such explanations were needed. The reviewer accepts these explanations and considers them appropriate. Thank you also for taking on board the suggestions for amendments. Now the work in the opinion of the reviewer can be published.